EMBO
Molecular Medicine

# LIMT is a novel metastasis inhibiting lncRNA suppressed by EGF and downregulated in aggressive breast cancer

Aldema Sas-Chen[1], Miriam R Aure[2,3], Limor Leibovich[4], Silvia Carvalho[1], Yehoshua Enuka[1], Cindy Körner[5], Maria Polycarpou-Schwarz[6,7], Sara Lavi[1], Nava Nevo[1], Yuri Kuznetsov[8], Justin Yuan[1], Francisco Azuaje[9], Oslo Breast Cancer Research Consortium (OSBREAC)[†], Igor Ulitsky[1], Sven Diederichs[6,7,10,11], Stefan Wiemann[5], Zohar Yakhini[4,12], Vessela N Kristensen[2,3], Anne-Lise Børresen-Dale[2,3] & Yosef Yarden[1,*]

## Abstract

Long noncoding RNAs (lncRNAs) are emerging as regulators of gene expression in pathogenesis, including cancer. Recently, lncRNAs have been implicated in progression of specific subtypes of breast cancer. One aggressive, basal-like subtype associates with increased EGFR signaling, while another, the HER2-enriched subtype, engages a kin of EGFR. Based on the premise that EGFR-regulated lncRNAs might control the aggressiveness of basal-like tumors, we identified multiple EGFR-inducible lncRNAs in basal-like normal cells and overlaid them with the transcriptomes of over 3,000 breast cancer patients. This led to the identification of 11 prognostic lncRNAs. Functional analyses of this group uncovered LINC01089 (here renamed LncRNA Inhibiting Metastasis; LIMT), a highly conserved lncRNA, which is depleted in basal-like and in HER2-positive tumors, and the low expression of which predicts poor patient prognosis. Interestingly, EGF rapidly downregulates LIMT expression by enhancing histone deacetylation at the respective promoter. We also find that LIMT inhibits extracellular matrix invasion of mammary cells *in vitro* and tumor metastasis *in vivo*. In conclusion, lncRNAs dynamically regulated by growth factors might act as novel drivers of cancer progression and serve as prognostic biomarkers.

**Keywords**  biomarkers; breast cancer; long noncoding RNA; migration; receptor tyrosine kinase
**Subject Category**  Cancer

## Introduction

Growth factors and their receptor tyrosine kinases (RTKs) play major roles in breast cancer progression (Hynes & Watson, 2010; Witsch *et al*, 2010). Of the five major breast cancer subtypes represented in the Prediction Analysis of Microarrays (PAM50) subtyping (Parker *et al*, 2009), the one overexpressing the RTK called HER2 and the less prevalent, the basal-like subtype are considered highly aggressive (Perou *et al*, 2000). While the former is driven by an amplified *HER2* gene, a fraction of the basal subtype is characterized by relatively high abundance of the epidermal growth factor receptor (EGFR, a kin of HER2) (Carey *et al*, 2010; Foulkes *et al*, 2010). In line with driver functions, EGFR-associated poor prognosis signatures are highly expressed in basal-like tumors, and blocking EGFR using either kinase inhibitors or monoclonal antibodies effectively retards growth of basal-like cancer cells (Hoadley *et al*, 2007; Ferraro *et al*, 2013).

Using basal-like untransformed cells, MCF10A, as a model system, data from our laboratory showed that both mRNAs and microRNAs exhibit dynamic changes in expression following EGF stimulation (Amit *et al*, 2007; Avraham *et al*, 2010; Kostler *et al*, 2013). We further demonstrated that the inducible mRNAs and

1  Department of Biological Regulation, Weizmann Institute of Science, Rehovot, Israel
2  Department of Cancer Genetics, Institute for Cancer Research, Oslo University Hospital, The Norwegian Radium Hospital, Oslo, Norway
3  K.G. Jebsen Centre for Breast Cancer Research, Institute for Clinical Medicine, University of Oslo, Oslo, Norway
4  Department of Computer Sciences, Technion–Israel Institute of Technology, Haifa, Israel
5  Division of Molecular Genome Analysis, German Cancer Research Center, Heidelberg, Germany
6  Division of RNA Biology & Cancer (B150), German Cancer Research Center (DKFZ), Heidelberg, Germany
7  Institute of Pathology, University Hospital Heidelberg, Heidelberg, Germany
8  Department of Veterinary Resources, Weizmann Institute of Science, Rehovot, Israel
9  Department of Oncology, Luxembourg Institute of Health, Luxembourg City, Luxembourg
10  German Cancer Consortium (DKTK), Freiburg, Germany
11  Division of Cancer Research, Department of Thoracic Surgery, Faculty of Medicine, Medical Center - University of Freiburg, University of Freiburg, Freiburg, Germany[‡]
12  Agilent Laboratories, Petach-Tikva, Israel
   *Corresponding author. Tel: +972 8 9343974; Fax: +972 8 9342488; E-mail: yosef.yarden@weizmann.ac.il
   [†]A list of members and their affiliations appears at the end of this article
   [‡]Correction added on 01 September 2016 after first online publication: Affiliations 7–11 have been reordered.

microRNAs are embedded into regulatory subnetworks, which are deregulated in diverse tumor types. Considering the emerging roles for long noncoding RNAs (lncRNAs) in metastasis of breast cancer (Serviss et al, 2014), we raised the possibility that some EGF-inducible lncRNAs might play a role in basal-like breast cancer. LncRNAs are transcripts greater than 200 nucleotides, which lack functional open-reading frames (Ponting et al, 2009; Rinn, 2014). They might regulate either local genomic regions (cis-regulation), which characterizes lncRNA ANRIL, or distant regions of the genome (transregulation), as in the case for lncRNA HOTAIR (Rinn et al, 2007). In addition, lncRNAs may act as scaffolds or as decoys. These interactions might regulate transcriptional mechanisms, including through epigenetic silencing (Gupta et al, 2010) or transcription activation (Orom et al, 2010). LncRNAs might also act post-transcriptionally, by sequestering microRNAs, or by controlling RNA processing and stability. This large functional diversity underlies the involvement of lncRNAs in a myriad of cellular processes, such as apoptosis (Hung et al, 2011) and metastasis (Gutschner et al, 2013).

In accordance with multiplicity of molecular targets, lncRNAs have been associated with several types of cancer (Gutschner & Diederichs, 2012; Niland et al, 2012), in which they might act as potential oncogenes or tumor-suppressor RNAs (Huarte & Rinn, 2010; Gibb et al, 2011). Moreover, because several lncRNAs can profoundly control transcription, their profiling might assist diagnosis, prognosis, or biomarker identification (Wang et al, 2011a,b). The involvement of lncRNAs in breast cancer progression is of particular interest (Shore et al, 2012). In the context of breast cancer, HOTAIR is upregulated in tumors (Gibb et al, 2011) and its overexpression might serve as an independent predictor of progression-free survival (Gupta et al, 2010). Similarly, LSINCT5, a polyadenylated stress-induced RNA, is overexpressed in breast cancer and affects cellular proliferation (Silva et al, 2011).

Since our model of EGF-stimulated mammary epithelial cells mirrors gene expression patterns in breast cancer patients, it has been employed herein with the aim of uncovering involvement of specific lncRNAs in progression of the basal-like subtype. To this end, we profiled EGF-induced changes in expression of lncRNAs and surveyed the prognostic value of individual, EGF-responsive genes. This led to the identification of a subset of eleven EGF-regulated lncRNAs, the expression patterns of which could be used to predict survival time of breast cancer patients. In vitro studies of the selected lncRNAs identified LINC01089/LIMT (LncRNA Inhibiting Metastasis), a hitherto uncharacterized EGF-downregulated lncRNA, as a regulator of mammary cell migration and invasion. Correspondingly, animal studies have shown that depletion of LIMT enhances metastasis formation in vivo. Importantly, we found that downregulation of LIMT characterizes breast cancer patients diagnosed with either basal-like or HER2-enriched tumors. Taken together, these results ascribe potential roles for inducible lncRNAs like LIMT in tumor progression.

# Results

## Expression levels of lncRNAs dynamically change upon stimulation of mammary cells with a growth factor

To uncover the potential involvement of lncRNAs in breast cancer metastasis, we first studied transcriptional responses of lncRNAs to EGF stimulation. MCF10A human mammary cells were treated with EGF for increasing time intervals (see scheme in Fig 1A). Purified RNA was then used to profile expression of both lncRNAs and mRNAs by means of gene expression microarrays. For each probe, we calculated the change in expression at every time point, relative to time zero (i.e. no EGF stimulation). Our initial survey noted that the overall dynamic range of lncRNAs was smaller than the dynamic range corresponding to mRNAs (i.e. log2 fold change ranged from $-3.67$ to $10.36$ for mRNAs and from $-2.32$ to $6.94$ for lncRNAs). Because of the narrower dynamic range of lncRNAs, transcripts were considered dynamic if their fold change was $> 1.5$ (lncRNAs) or 2 (mRNAs) in at least one time point relative to time zero. Accordingly, the expression of 346 lncRNAs was affected by EGF (Fig 1B, left panel). Clustering the responsive lncRNAs according to peak expression times identified waves of transcription, similar to those observed with mRNAs (Fig 1B, right panel) and with miRNAs (Avraham et al, 2010). Unlike mRNA, the majority of lncRNAs exhibited downregulation in response to EGF stimulation. However, similar to mRNAs, many lncRNAs displayed very rapid responses; their down- or upregulation initiated as early as 20 min after stimulation (confirmed by use of quantitative PCR; Fig EV1). These early events are of special interest as they might represent immediate responses to EGF signaling, rather than later, indirect effects of the signaling cascade. In conclusion, the abundance of a group of lncRNAs dynamically changes following short treatments of mammary cells with a growth factor.

## The abundance of EGF-regulated lncRNAs predicts clinical outcome of breast cancer patients

Next, we assessed clinical cohorts for expression of the EGF-regulated lncRNAs from MCF10A cells. Two datasets of breast cancer patients were analyzed: The METABRIC dataset (Curtis et al, 2012) and several breast cancer cohorts integrated into a single dataset, called "Kaplan–Meier (KM) plotter" (Gyorffy et al, 2010). Each dataset includes ~2,000 breast cancer patients, who were followed for $> 20$ years from initial diagnosis. In addition to gene expression data, derived from either Illumina (Curtis et al, 2012) or Affymetrix (Gyorffy et al, 2010) hybridization arrays, each dataset also includes information on relapse-free and overall patient survival time. A Kaplan–Meier (KM) survival curve was generated for each of the EGF-responsive lncRNAs. This analysis identified eleven EGF-regulated lncRNAs, the expression of which was found to be significantly associated with patients' survival (adjusted P-value $< 0.05$), in at least one dataset (Fig EV2). Figure 2A presents the kinetics of abundance alterations of two such early response lncRNAs, LIMT and LOC388796. Interestingly, while low expression of the EGF-downregulated lncRNA called LIMT predicted shorter overall and relapse-free patient survival (Fig 2B), high expression of LOC388796, which is upregulated in response to EGF, predicted poor patient prognosis (Fig EV2).

Although some lncRNAs encode short peptides, and they might occupy ribosomes, most lncRNAs possess significantly lower coding potentials as compared to protein-coding genes (Dinger et al, 2008; Guttman et al, 2013; Ruiz-Orera et al, 2014). In order to verify that the eleven EGF-regulated transcripts identified above were indeed noncoding, we examined their coding probability using the Coding-Potential Assessment Tool—CPAT (Wang et al, 2013). As reference,

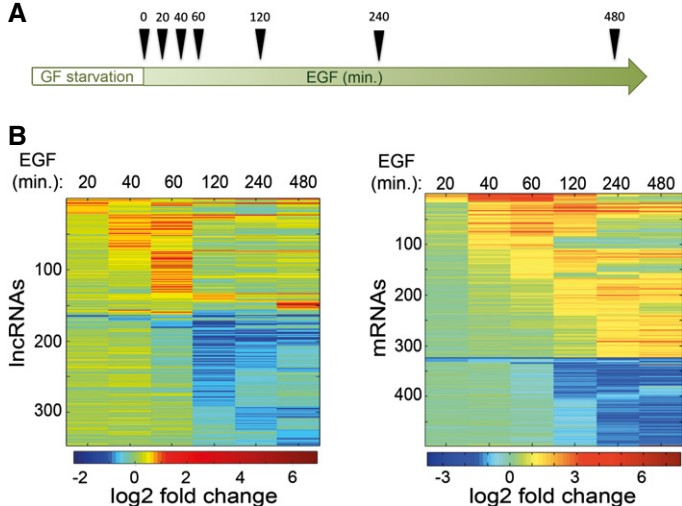

**Figure 1.  EGF stimulation instigates dynamic changes in expression of multiple lncRNAs.**

A   A scheme of the experimental design. Triangles represent the time points (in minutes) of cell harvesting for RNA purification, after EGF stimulation (GF, growth factor).

B   MCF10A cells were stimulated with EGF (10 ng/ml) for up to 8 h. At the indicated time points, cells were harvested, and RNA was isolated and used to determine expression levels of specific lncRNAs and mRNAs, using the SurePrint microarray platform (from Agilent). The heatmaps depict lncRNAs (left) and mRNAs (right) for which a > 1.5-fold (lncRNAs) or a > twofold (mRNA) change in expression was observed in at least one of the time points, compared to unstimulated cells (time zero). Note that each row of the heatmap represents an individual lncRNAs/mRNAs, and values (see color scale at the bottom) indicate the log2 ratio of RNA levels between each time point and time zero.

we used coding potentials of 4,000 protein-coding RNAs and 4,000 known lncRNAs. Interestingly, the coding probabilities of all eleven lncRNAs we identified were similar to those of known lncRNAs and significantly lower than those of coding RNAs (Fig 2C). In addition, the evolutionary conservation of the 11 genes displayed wide variation, with LIMT being the most conserved (Fig 2D). In summary, we identified several noncoding RNAs, some of which are evolutionarily conserved. All these lncRNAs undergo abundance alterations in response to short treatments of human mammary cells with EGF, which might predict disease aggressiveness and patient outcome.

## LIMT, an EGF-downregulated lncRNA, acts as an inhibitor of motility *in vitro* and retards metastasis in an animal model

Knocking down the expression of specific lncRNAs might uncover their cellular functions. Hence, we transfected MCF10A cells with lncRNA-specific siRNAs (available for nine out of the 11 clinically relevant lncRNAs) and measured knockdown efficiency using qPCR (Fig EV3A). Because efficient knockdown was achieved for LINC01089/LIMT, LOC642852, LOC344595, and LOC282997, the cellular functions of these genes were addressed using apoptosis and viability assays (Fig EV3B). Transfection with polo-like kinase 1 (PLK1)-specific siRNAs was used as reference, since knockdown of PLK1 usually leads to extensive apoptosis. In terms of viability and apoptosis, the only effect we observed was a slight increase in the fraction of dying cells following knockdown of LOC282997 (aka PDCD4-AS1, Fig EV3B), implying that the examined lncRNAs are not involved in cell survival or apoptosis.

Since MCF10A cells adopt migratory phenotypes in response to EGF-induced stimulation (Tarcic *et al*, 2012), we examined the effect of lncRNA knockdown on cellular motility. Firstly, we placed siRNA-transfected cells in the upper compartment of Transwell migration chambers and determined their migration to the lower compartment, 20 h later, in the presence of EGF. In this assay, EGFR-specific siRNAs were used as reference, and indeed, we observed strong inhibition of cellular migration following *EGFR* knockdown (86%; Fig EV3C). Knockdown of only one lncRNA, LIMT, caused a similarly large, but opposite effect on migration (Figs 3A and EV3C). Consistent with this observation, knockdown of LIMT also increased the capacity of cells to invade through a layer of extracellular matrix (Fig 3B). To complement this loss-of-function approach, we created an MCF10A subline stably overexpressing LIMT (or eGFP as control; see Fig EV3D). As expected, ectopic expression of LIMT reduced migratory and invasive capacities (Fig 3A and B). To further validate the effect of LIMT on migration, we designed two shRNAs directed against different parts of the gene and show results for sh716, because it could more effectively downregulate the respective transcript (Fig EV3E, see Materials and Methods). Consistent with the observations made with siRNAs and with overexpression of LIMT, MCF10A cells stably expressing shLIMT displayed remarkably increased migration and invasion relative to control cells (Fig 3A and B).

Taken together with the ability of EGF to decrease expression of LIMT (Fig 2A), the migration and invasion results proposed the following scenario: EGF treatment downregulates LIMT, an inhibitor of cell migration and invasion, thereby enhancing motility of mammary cells. To test this model, we used a chemical inhibitor, U0126, which specifically blocks a major signaling pathway downstream of EGFR, the RAS-to-ERK pathway. We first confirmed blocking efficacy by assaying ERK phosphorylation

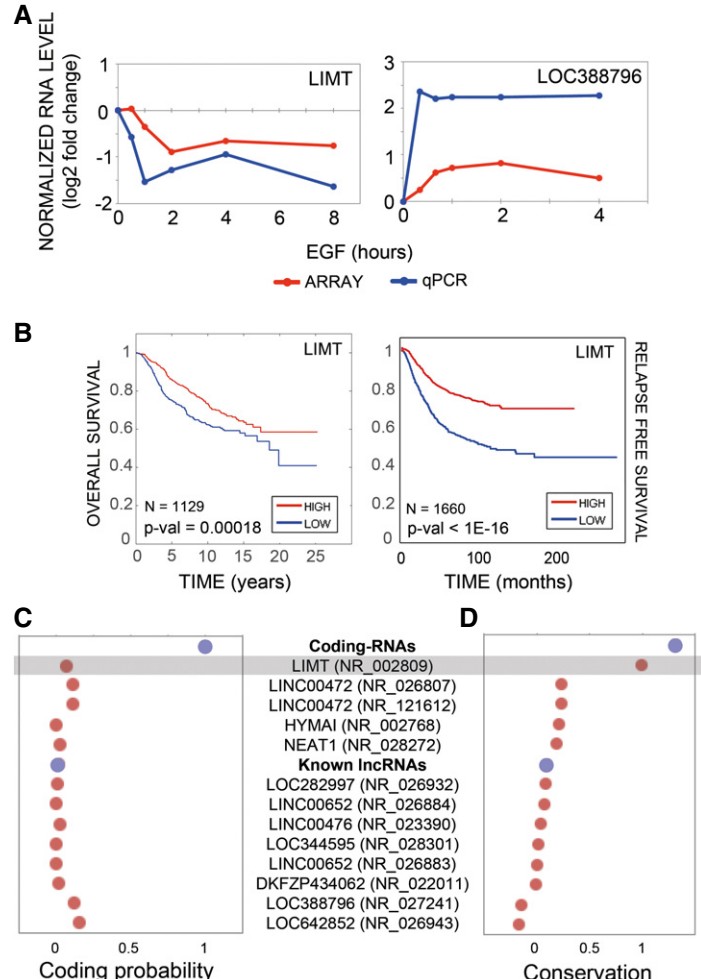

**Figure 2. Changes in abundance of EGF-regulated lncRNAs correlate with clinical outcome of breast cancer patients.**

A  Shown are levels of two EGF-regulated lncRNAs: the EGF-downregulated lncRNA called LIMT (left) and the upregulated lncRNA called LOC388796 (right). RNA abundance was determined using microarrays (red) and real-time qPCR (blue) and presented as fold change relative to time zero. Beta-2-microglobulin was used for normalization.

B  Shown are Kaplan–Meier plots of overall survival and relapse-free survival for the EGF-regulated lncRNA called LIMT. To obtain the data, we overlapped the list of EGF-regulated lncRNAs with the METABRIC clinical dataset (Illumina platform; left panel) and the KM-plotter dataset (Affymetrix platform; right panel). The red and blue lines of the left panel correspond to high and low expressors, respectively (each shows one-third of the population; 1129 out of a total of 1,693 patients). The same applies to the right panel, except that the population was divided into two equal size groups (N = 1,660 patients).

C  Coding potentials of clinically significant lncRNAs were calculated using CPAT, and they are individually presented along with the mean coding probabilities of control groups of protein-coding RNAs and lncRNAs (in blue). Note that two variants of LINC00472 and LINC00652 are presented.

D  Evolutionary conservation of the primary nucleotide sequences of the clinically significant EGF-regulated lncRNAs was calculated using PhyloP across 100 vertebrates. Individual conservation scores are presented along with that of control groups of protein-coding RNAs and noncoding RNAs (in blue).

(Fig 3C), as well as by measuring mRNA levels of EGR1, a downstream target of the RAS-to-ERK pathway (Fig EV3F). Ultimately, we measured RNA levels of LIMT and found that inhibiting the ERK pathway, using the MEK inhibitor, negated the decrease in abundance of LIMT (Fig 3D). This observation suggests that the RAS-to-ERK pathway mediates EGF-induced downregulation of LIMT, as well as the consequent enhancement of cellular motility.

Since high abundance of LIMT is associated with longer survival of advanced state breast cancer patients (Fig 2B and Appendix Fig S1), we tested the prediction that overexpression of LIMT would attenuate metastasis *in vivo*. For this, we established a derivative of the highly metastatic MDA-MB-231 triple negative breast cancer cell line, which stably expresses an ectopic LIMT (or eGFP). As a prelude to the metastasis assays, we validated that manipulating the abundance of LIMT in MDA-MB-231 can imitate the migration effects we observed with MCF10A cells (Fig EV4). Next, we injected RFP-labeled MDA-MB-231 cells overexpressing LIMT (or eGFP) into the tail vein of female SCID mice. Eighteen days after injection, lungs were excised and the number of metastatic nodules was quantified. Consistent with the *in vitro* results, overexpression of LIMT reduced the number of detectable metastatic nodules (Fig 4A), supporting a role for LIMT in inhibiting metastasis formation *in vivo*. To corroborate these tests, we

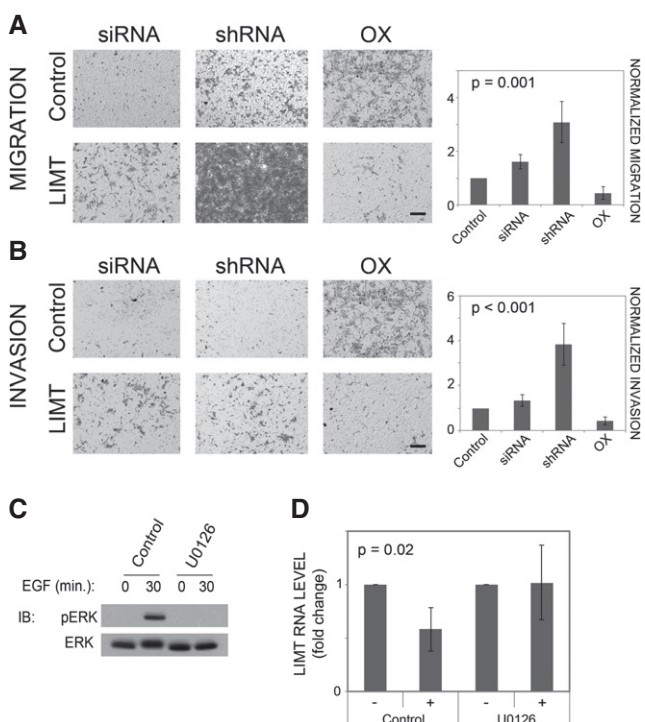

**Figure 3. The ERK pathway mediates the effect of EGF on LIMT, which normally inhibits mammary cell migration and invasion.**

A, B MCF10A cells were transfected with LIMT-specific siRNAs (or with control siRNA), and their migration (A, left panels) or invasion (B, left panels) were determined. Likewise, cells were stably transfected with plasmids encoding for shRNAs against LIMT (middle panels) or with plasmid encoding for either LIMT or eGFP (Control). Also shown is quantification of cell migration and invasion upon lncRNA knockdown and overexpression (OX). *P*-values of one-way ANOVA are presented. Each assay was repeated at least three times.

C, D MCF10A cells were treated for 30 min with U0126 (a MEK inhibitor), or with no agent, and thereafter, EGF was added (10 ng/ml) and incubated with cells for additional 30 min (C) or 4 h (D). Protein extracts were used to assess levels of phosphorylated ERK (C). Isolated RNA was used to determine abundance of LIMT using real-time qPCR (D). Expression values are presented as fold change relative to time zero. Beta-2-microglobulin was used for normalization. *P*-value of *t*-test from four repeats is shown. − : no EGF, + : EGF for 4 h.

Data information: All values represent mean ± SD of replicates.
Source data are available online for this figure.

employed a reciprocal approach utilizing MDA-MB-231 cells stably expressing the selected shRNA specific to LIMT. As predicted, cells stably expressing shLIMT (sh716) displayed relatively high migratory capacity *in vitro* (Fig EV4). Hence, both knocked-down cells and control cells were then subjected to the above-described *in vivo* metastasis assay. Analyses of fluorescence images of the lungs from the LIMT knockdown group confirmed the ability of LIMT to inhibit metastasis of mammary cells from the circulation to lungs (Fig 4B). Overall, two independent lines of animal studies attributed to this EGF-downregulated lncRNA an important role in regulating metastasis formation; hence, we denoted it as LIMT (LncRNA Inhibiting Metastasis).

**The 5′ region of *LIMT* is highly conserved in vertebrates and undergoes histone deacetylation in response to EGF**

To gain deeper understanding of LIMT, we explored its genomic characteristics. *LIMT* maps to chromosome 12, between the protein-coding genes *SETD1B* and *RHOF*. In accordance with earlier reports (Ulitsky *et al*, 2011), the human *LIMT* gene is highly conserved throughout most of its length, with its 5′ portion

(which encodes the first four exons) showing a very high degree of conservation in vertebrates (Fig 5A, green rectangle). This observation hints that a cellular function of LIMT, such as recognition of other molecules, might localize to the transcript's 5′ region. Because LIMT is dynamically expressed in mammary cells, we predicted the presence of histone modifications at the 5′ region. Analyses of histone marks characteristic of transcriptionally active, open chromatin structures, such as H3K4Me3, in the vicinity of the transcription start site of *LIMT*, were consistent with an active promoter (Fig 5A). To corroborate this observation, we investigated changes in tri-acetylation of histone 3 (lysine 27; H3K27Ac), a marker of transcriptional activity. This was done by immunoprecipitating H3K27Ac from MCF10A cells, after stimulation with EGF. Profiling immunoprecipitated DNA fragments using deep sequencing detected an overall decrease in acetylation of histone 3 at lysine 27, especially at the 5′ region, as early as 20 min post-stimulation (Fig 5A, red rectangle). This observation is consistent with the observed EGF-induced reduction in transcript abundance (Fig 2A). Interestingly, although some lncRNAs have been shown to affect the expression of neighboring genes in cis, the abundance of LIMT showed only

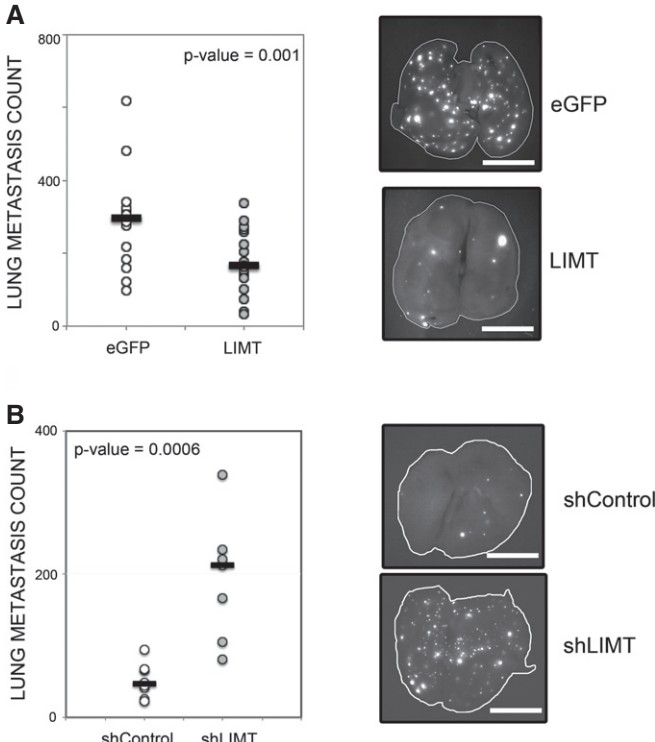

**Figure 4.  LIMT inhibits metastasis formation *in vivo*.**

A  RFP-labeled MDA-MB-231 cells stably overexpressing LIMT or eGFP were injected into the tail vein of 5-week-old female SCID mice (150,000 cells/mouse). Eighteen days after injection, lungs were excised and imaged. The number of metastatic nodules in each lung was quantified and presented in a dot plot. The horizontal lines represent median number of nodules per animal of each group. Each dot of the left panel represents one animal. The experiment was repeated twice ($N = 18$ and 14 for LIMT and eGFP overexpression, respectively). Shown are representative fluorescence images of lungs from the LIMT overexpression group and the control group. Scale bar, 0.5 cm. A two-way ANOVA was applied to evaluate differences between groups.

B  The experiment described above was carried out with RFP-labeled MDA-MB-231 cells stably expressing shRNAs against LIMT ($N = 7$) or control shRNAs ($N = 8$). Shown are representative fluorescence images of lungs from the LIMT knockdown group and the control group. Scale bar, 0.5 cm. A Student's *t*-test was applied to evaluate differences between groups. The experiment was repeated with a second shRNA and yielded similar results.

weak correlation with transcript levels of the flanking genes, namely SETD1B and RHOF (Fig EV5A and B). This observation suggests that expression of LIMT is uncoupled from that of the neighboring, protein-coding genes.

### Identification of candidate LIMT-regulated genes

Because lncRNAs might act as scaffolds or decoys, which interact with microRNAs (Poliseno *et al*, 2010; Yuan *et al*, 2014) and the epigenetic machinery (Rinn, 2014) to regulate transcription, we explored potential downstream targets of LIMT. To this end, we firstly manipulated the levels of LIMT in MCF10A cells, by means of either siRNA-mediated knockdown or stable overexpression. We then profiled genomewide abundance of RNAs using Affymetrix microarrays. Analysis of the microarray screens identified 48 genes that undergo reciprocal expression changes in response to LIMT knockdown and overexpression (i.e. at least a 1.5-fold change in expression compared to control; Fig EV5C). Remarkably, the selected group of genes included several noncoding RNAs and coding genes, such as *TGFB2*, *SERPINB2*, *PTPRZ1*, *RHOB* and *IL-24*, which are known to be regulators of cellular migration. Future studies will test direct or indirect interactions

between LIMT and the respective promoters, or their chromatin marks.

### LIMT displays wide tissue distribution and reduced expression in relatively aggressive breast tumors

The finding that LIMT is downregulated in response to EGFR signaling and the ability of LIMT to inhibit metastasis in animal models correspond to our observations associating reduced LIMT expression with poor prognosis of breast cancer patients (Fig 2B). This corollary led us to explore LIMT's expression patterns. As a first step, we analyzed a panel of 44 cell lines representing cancer and non-cancer cells from 15 tissues of human origin, including six breast cell lines. LIMT was found to be widely and differentially expressed, with relatively high expression in brain, blood, and several lung and mammary cell lines (Fig 5B).

Next, we evaluated relations between LIMT and molecular subtypes of breast cancer, in two cohorts of patients: the Oslo2 cohort (Aure *et al*, 2014) and the larger, METABRIC dataset (Curtis *et al*, 2012). In line with the aforementioned lines of evidence, the lowest expression of LIMT in both cohorts corresponded

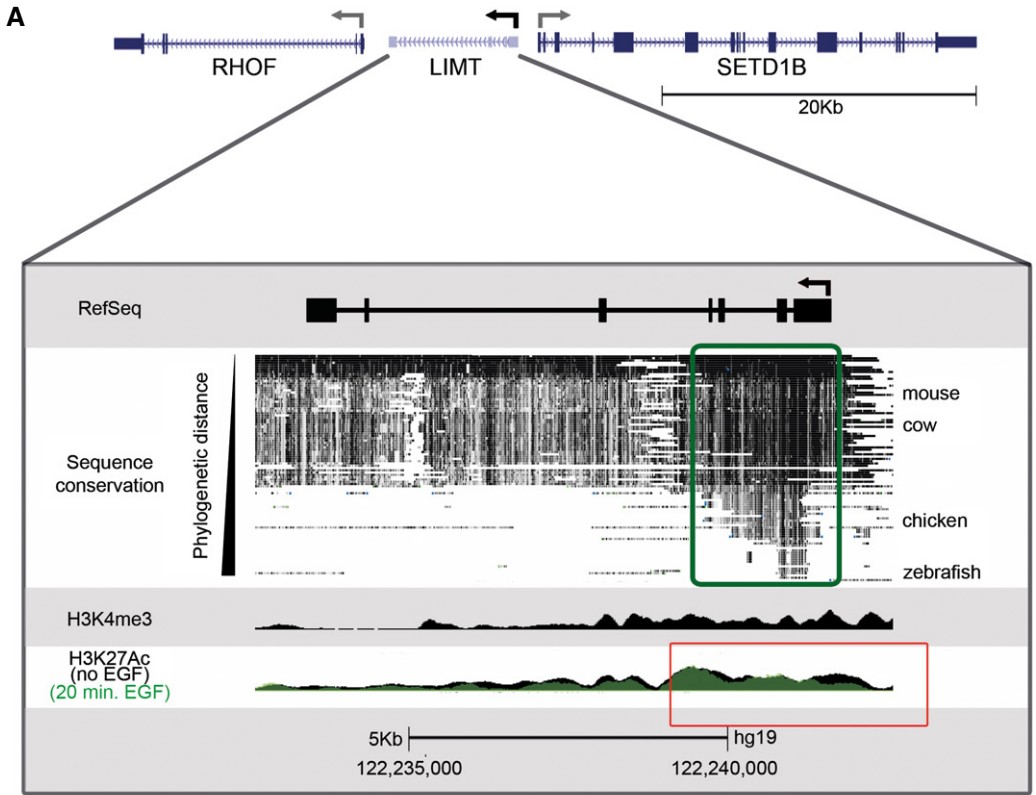

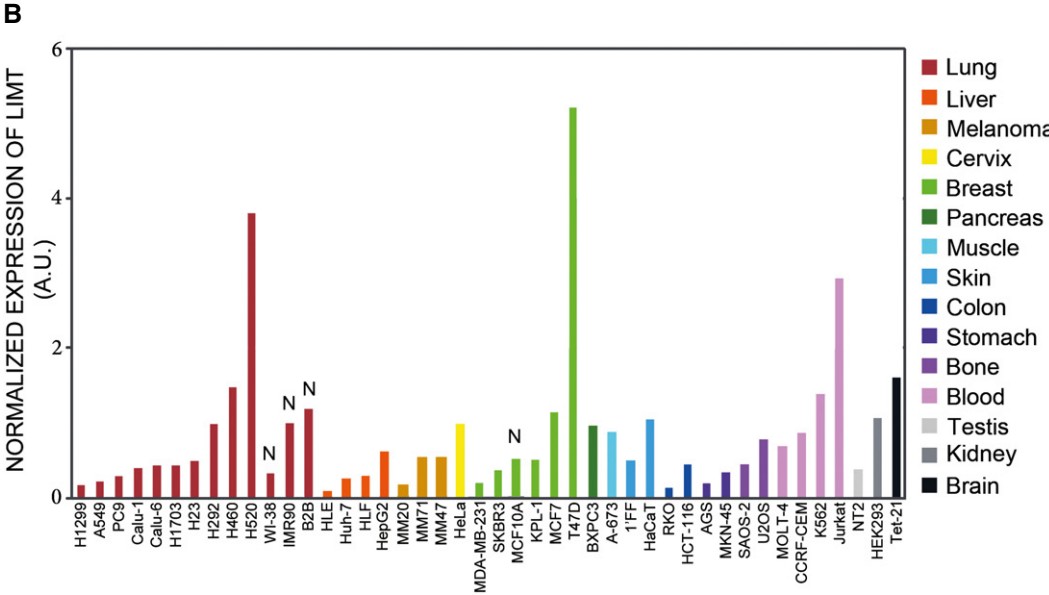

**Figure 5. Features of the *LIMT* gene and RNA abundance in cell lines of different tissues of origin.**

A Schematic representation of the annotated RefSeq gene models of *LIMT* (NR_002809; chromosome 12q24.31) and the neighboring protein-coding genes (*RHOF* and *SETD1B*). Arrows denote direction of transcription (note that *LIMT* resides on the minus strand). Also shown is the distribution of evolutionary conservation along the sequence of *LIMT* across 100 vertebrate species. Note that the 5′ region of *LIMT* is highly conserved (green rectangle). Promoter-associated histone methylation (H3K4Me3) and acetylation (H3K27Ac3) were obtained in MCF7 and MCF10A cells, respectively (see Materials and Methods). Acetylation signals were obtained for MCF10A cells stimulated with EGF for 20 min (green), or unstimulated (black; overlaying histogram). Note that stimulation with EGF caused a decrease in acetylation at the promoter region (red rectangle).

B Expression of *LIMT* was determined in a panel of 44 cancer and non-cancer cell lines using real-time qPCR. Expression levels were normalized to the GAPDH transcript and presented in arbitrary units (A.U.). N, non-cancer cell line.

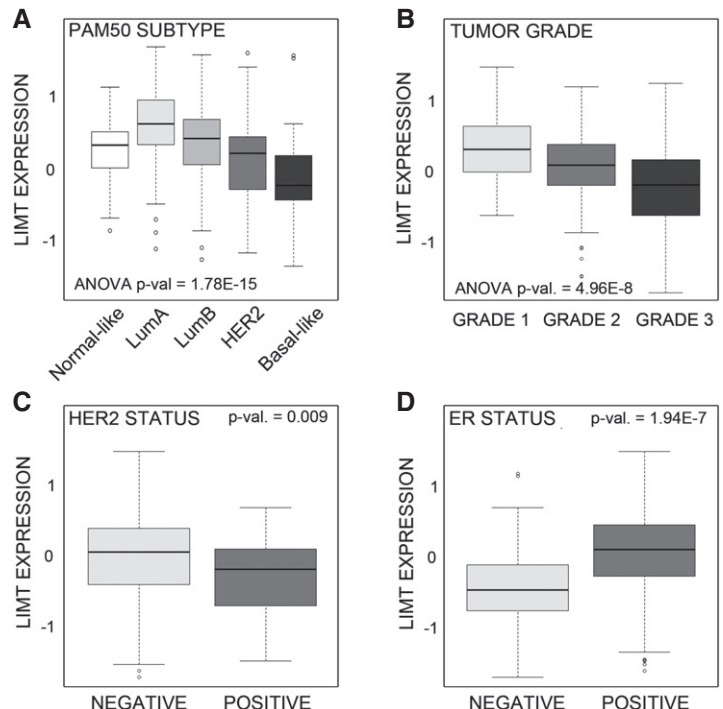

**Figure 6.  Expression of *LIMT* in breast cancer specimens associates with disease parameters.**

A–D   The expression of *LIMT* is shown relative to (A) PAM50 subtypes (*N* = 381 patients), (B) tumor grade (*N* = 309), (C) HER2 status (*N* = 309), and (D) ER status (*N* = 381 patients). Expression of the lncRNA was determined in tumors from breast cancer patients of the Oslo2 study using Agilent arrays. Molecular subtypes of the corresponding tumors were determined by using the PAM50 classifier. The Student's *t*-test was applied to evaluate differences in lncRNA expression between two groups. To evaluate differences in expression among three or more groups, we applied one-way analysis of variance (ANOVA). The bottom and top of the box represent the first and third quartiles of the data, respectively, and the band inside the box represents the median of the data. The lower and upper whiskers represent the lowest and highest data points of the data, respectively. Circles represent outliers, defined as samples deviating by more than 1.5× interquartile range (IQR) from the upper or lower quartiles, respectively.

to patients diagnosed with basal-like cancer (Fig 6A and Appendix Fig S1), an aggressive tumor frequently associated with high EGFR expression and mutations in the downstream pathways (Hoadley *et al*, 2007; Foulkes *et al*, 2010). Similarly, low expression of LIMT was noted in the other aggressive subgroup, HER2 enriched. In line with reduced expression in relatively aggressive tumors, low LIMT expression correlated with higher tumor grade and stage, HER2 positivity and ER negativity (Fig 6B–D and Appendix Fig S1). In conclusion and in line with the analyses of patient survival, we performed on additional cohorts, the loss of LIMT might coincide with the acquisition of aggressive features by breast cancers.

In summary, by investigating a model system that simulates autocrine and stromal mechanisms controlling invasion of mammary tumor cells across tissue barriers, we identified several lncRNA molecules, the expression of which is altered in response to a growth factor and might predict disease course. Focusing on one of these, LIMT (LINC01089), we obtained evidence in support of an EGF-induced, ERK-mediated downregulation of this noncoding RNA molecule. Manipulating LIMT's transcript levels inhibited the ability of mammary cells to migrate *in vitro*, as well as altered their ability to form metastases in mice. In line with clinical relevance of the EGFR–ERK–LIMT regulatory module, reduced expression of LIMT marks mammary tumors of patients diagnosed with relatively

aggressive and advanced forms of the disease. Below, we discuss potential mechanisms and clinical implications of LIMT and other growth factor-responsive lncRNAs.

# Discussion

Breast cancer is a heterogeneous disease; hence, many efforts are being made to identify new drivers and classify tumors into increasing numbers of subtypes (Perou *et al*, 2000; Dawson *et al*, 2013). Thus, whereas initial profiling focused on finding histological characteristics and signatures of protein-coding genes, more recent studies have found that the expression of microRNAs may also assist prognosis and classification (Dvinge *et al*, 2013). Importantly, the relatively high tissue specificity of lncRNAs promises a role for these noncoding RNAs as drivers and biomarkers of breast cancer (Wang *et al*, 2011a). For example, a recent survey of 658 infiltrating ductal tumors reported that the lncRNA HOTAIR was significantly overexpressed in the HER2-positive subgroup, while the lncRNA HOTAIRM1 was significantly overexpressed in the basal-like subgroup (Su *et al*, 2014).

Herein, we introduce a novel approach based on the identification of inducible lncRNAs. Our focus on growth factor-responsive lncRNAs was based on the fundamental involvement of

stroma and growth factors in multiple steps of tumor progression (Witsch *et al*, 2010), including progression of basal-like tumors (Foulkes *et al*, 2010). Hence, expression of the EGF-inducible lncRNAs we uncovered was examined in large cohorts of breast cancer patients, with the vision of identifying potential drivers and biomarkers of the basal-like and the HER2-enriched subtypes. The unbiased screen identified several EGF-responsive lncRNAs, which have previously been associated with cancer progression, demonstrating the suitability of our approach. For example, Neat1, which is overexpressed in advanced ovarian carcinoma and assists diagnosis (Pils *et al*, 2013), and LINC00472, high expression of which is correlated with reduced risk of relapse and death of breast cancer patients (Shen *et al*, 2015). Importantly, both lncRNAs were found to be early responders to EGF in our cellular model.

Notably, the analysis we performed also identified new candidate lncRNAs, with potential roles in breast cancer progression and profiling. Functional assays of the candidates, which made use of RNA interference, led us to focus on a previously uncharacterized lncRNA, referred herein as LIMT (originally denoted as LINC01089 or LOC338799). We found that downregulation of this lncRNA depends on prior activation of EGFR and the RAS-to-ERK pathway. Notably, this signaling route is frequently activated in hormone-independent breast cancer and serves as a driver of mammary cell migration (Tarcic *et al*, 2012). Accordingly, we found that knockdown of LIMT enhanced cellular migration and invasion *in vitro*, as well as metastasis *in vivo*. Because lncRNAs such as ANRIL, SRG1 and Paupar might regulate in cis the expression of nearby genes (Martens *et al*, 2004; Kotake *et al*, 2011), we investigated whether *LIMT*'s neighbors, namely the protein-coding genes *RHOF* (positioned 1,579-bp downstream of *LIMT*) and *SETD1B* (located on the opposite strand to *LIMT*, at a distance of 1,248 bp), are co-expressed in tumors. However, expression of these genes did not correlate to that of LIMT.

Understanding the mechanisms by which LIMT regulates cellular migration and tumor progression is a matter of further investigation. Because LIMT, and especially its 5′ region, is one of only 29 lincRNAs that display very high sequence conservation throughout all vertebrates (Ulitsky *et al*, 2011), we assume that LIMT's function is shared by several species and that it requires a specific structure at the 5′ region. Presumably, physical interactions of LIMT with protein or RNA molecules regulate transcription of coding and noncoding genes involved in cellular motility. Some of these genes may appear in the groups we identified by applying array technology on LIMT-manipulated mammary cells. Alternatively, like several other lncRNAs, LIMT might interact with the epigenetic machinery (Rinn, 2014). Current studies also uncover binding of lncRNAs with microRNAs (Poliseno *et al*, 2010; Yuan *et al*, 2014), and yet other studies unearth roles of peptides encoded by short open-reading frames present within putative lncRNAs (Bazzini *et al*, 2014; Ruiz-Orera *et al*, 2014). Regardless of the underlying mechanism, reduced expression of LIMT emerges from the present study as a trait of relatively aggressive, basal-like and HER2-driven breast tumors. In the same vein, we found that LIMT is downregulated in grade 3 and in stage 4 mammary tumors compared to less advanced tumors. Thus, LIMT might represent a new class of EGF-controlled and ERK-mediated inhibitors of breast cancer metastasis, which function as tumor-suppressor lncRNAs.

## Materials and Methods

### Materials and cell lines

MCF10A cells were grown as described (Tarcic *et al*, 2012). For time course experiments, cells were starved overnight in DME:F12 (1:1) medium without additives (starvation medium) and at the time of stimulation, EGF was added to a final concentration of 10 ng/ml. MDA-MB-231 cells were grown in DME medium supplemented with 10% fetal calf serum. The listed 44 normal and cancer cell lines were cultured in their respective media and supplements (Appendix Table S1). An anti-ERK2 rabbit polyclonal antibody was purchased from Santa Cruz Biotechnology. The anti-phosphorylated ERK1/2 antibody was purchased from Promega. The goat anti-rabbit IgG conjugated to horseradish peroxidase (HRP) was purchased from Jackson ImmnoResearch Laboratories. A rabbit polyclonal antibody to the acetylated form (lysine 27) of histone 3 was purchased from Abcam (ab4729).

### RNA purification and real-time quantitative PCR

RNA isolation was performed using the TRIzol reagent (Life Technologies) or the PerfectPure RNA Cultured Cells kit (5 Prime). Generation of cDNA was performed using either qScript cDNA synthesis kit (Quanta), High-capacity cDNA reverse transcription kit (Applied Biosystems) or RevertAid Reverse Transcriptase (Thermo Scientific). Real-time qPCR analysis was performed using Fast SYBR Green Master Mix (Applied Biosystems). Primers were designed using PrimerBlast, and their sequences appear in Appendix Table S2. Transcripts encoding beta-2 microglobulin (B2M) and glyceraldehyde-3-phosphate dehydrogenase (GAPDH) were used for normalization.

### Lysate preparation and immunoblotting analyses

Cells were harvested in solubilization buffer (50 mM HEPES (pH 7.5), 150 mM NaCl, 10% glycerol, 1% Triton X-100, 1 mM EDTA, 1 mM EGTA, 10 mM NaF, 30 mM beta-glycerol phosphate, 0.2 mM $Na_3VO_4$, and a protease inhibitor cocktail). Heated (95°C) lysates were loaded onto acrylamide gels and resolved using electrophoresis, followed by electrophoretic transfer to a nitrocellulose membrane. After transfer, nitrocellulose membranes were blocked in TBST buffer (0.02 M Tris–HCl (pH 7.5), 0.15 M NaCl, and 0.05% Tween-20) containing 5% low-fat milk, blotted overnight using a primary antibody, washed in TBST, and incubated for 30 min with a secondary antibody linked to HRP.

### siRNA oligonucleotides and transfection methods

The siRNA oligonucleotide was purchased from Dharmacon. For knockdown experiments, cells were seeded in 6-well plates at 50% confluence, and 24 h later, they were transfected with siRNAs (20–50 nM) using the Lipofectamine 2000 transfection reagent (Life Technologies). Forty-eight hours later, cells were harvested and RNA was purified in order to verify knockdown efficiency using real-time qPCR. Sequences of siRNA oligonucleotides are presented in Appendix Table S3.

    

## Plasmids and infections

For overexpression of LIMT, the nucleotide sequence of LINC01089/LIMT (NR_002809) was purchased from BlueHeron and cloned into a pLEX_307 expression vector. As control, eGFP was cloned into pLEX_307. For knockdown of LIMT: Sequences targeting LIMT were designed using siRNA Wizard (http://www.invivogen.com/sirnawizard) and the GPP site (http://www.broadinstitute.org/rnai/public), as well as sequences designed by Dharmacon for siRNAs. Each shRNA sequence was cloned into a pLKO.1 plasmid (Addgene #10878). An shControl pLKO.1 plasmid was used as control (Addgene #10879). Sequences of shRNAs appear in Appendix Table S3. Lentiviruses encapsulating the *LIMT* expression vector or the shRNA expression vectors (along with the respective controls) were produced in HEK-293FT cells and used to infect MCF10A or RFP-labeled MDA-MB-231 cells. Selection of cells was done using puromycin.

## Cell migration and invasion assays

Forty-eight hours after transfection with siRNAs, cells were counted and reseeded on the upper face of Transwell migration or invasion chambers (Thermo Scientific). Cells (40,000–120,000 per chamber) were seeded in full medium and left at 37°C for 20 h. The same number of cells was seeded in parallel in 12-well plates and used as control for seeding variation. Twenty hours later, cells were fixed in paraformaldehyde (3% in saline), washed, and stained, using crystal violet. Cells attached to the upper face of the chamber were removed, and only the remaining migrating cells were imaged using a binocular. ImageJ was used for quantification of migration and invasion results.

## Cell viability assays

MCF10A cells were seeded in 96-well plastic-bottom plates (Greiner) at 50% confluence, and 24 h later, they were transfected with siRNAs (30 nM) in six replicates, using Lipofectamine 2000 (Life Technologies). Seventy-two hours later, cells were treated for 3 h with the WST-1 reagent (Roche) according to the manufacturer's instructions. Absorbance (at 450 nm) was assessed using a Tecan plate reader.

## Apoptosis assays

MCF10A cells were seeded in 96-well glass bottom plates (Matrical) at 50% confluence, and 24 h later, they were transfected with siRNAs in six replicates (30 nM) using the Lipofectamine 2000 transfection reagent (Life Technologies). Forty-eight hours later, cells were stained with Hoechst-33258 (Sigma) and apoptotic cells were stained for activated caspase 3/7 (for 60 min) using NucView-488 (Biotium). Plates were imaged using an automated ScanR screening microscope (Olympus). Nuclei and the fraction of apoptotic cells were counted using the ScanR software.

## Gene expression microarrays

Fifteen RNA samples, representing biological duplicates of MCF10A cells stimulated with EGF (10 ng/ml) for 20, 40, 60, 120, 240, and 480 min, as well as triplicates of non-stimulated cells, were analyzed using SurePrint G3 Human GE 8×60K one-color microarrays (Agilent Technologies). The raw data were normalized and filtered in order to make the experiments comparable with each other and in order to reduce noise. Furthermore, probes having low-quality measurements were filtered out. Next, for each probe, the log2 ratio of expression between each time point and time "zero" was calculated. Genes were considered changing if a fold change of > 0.6 log2 was observed in at least one time point. Additionally, since many of the lncRNA probes were based on putative lncRNA annotations, we overlapped the position of each probe with RefSeq annotated ncRNAs. Only probes overlapping annotated RefSeq ncRNAs were retained for assessment of the effect of EGF stimulation on transcript abundance. The microarray dataset is accessible at the ArrayExpress database (E-MTAB-4822). In addition, we used DNA arrays to profile gene expression upon manipulation of LIMT levels. Twelve RNA samples, representing (i) knockdown of LIMT, (ii) knockdown using control siRNA (iii) overexpression of LIMT, and (iv) overexpression of eGFP (all in biological triplicates), were analyzed using GeneChip® Human Gene 2.0 ST Arrays (Affymetrix). The raw data were normalized and used to identify genes differentially affected by knockdown versus overexpression of LIMT in MCF10A cells. The microarray dataset is accessible at the ArrayExpress database (E-MTAB-4821).

## Histone acetylation analysis

Cell fixation, harvest, chromatin immunoprecipitation, library generation, and sequencing were performed as described (Blecher-Gonen *et al*, 2013). For immunoprecipitation, we used a rabbit polyclonal antibody (ab4729 from Abcam) specific to the acetylated form (lysine 27) of histone 3. ChIP-seq reads were mapped to the hg37 genome assembly using bowtie2 (Langmead & Salzberg, 2012). Data presentation used the MACS2 program and fragment size of 270.

## Assessment of coding probability

Coding potential was calculated using an alignment-free method, Coding-Potential Assessment Tool (CPAT), which recognizes coding and noncoding transcripts from a large pool of candidates (Wang *et al*, 2013). Alongside, we used a reference catalog of human genes, which lists multiple protein-coding sequences ($N = 4,000$) and lncRNAs ($N = 4,000$) (Cabili *et al*, 2011).

## Assessment of the evolutionary conservation of lncRNAs

Evolutionary conservation of the primary nucleotide sequences of the clinically significant EGF-regulated lncRNAs was calculated by averaging PhyloP conservation scores (Pollard *et al*, 2010) of the exonic regions of each lncRNA using the bigWigAverageOverBED utility (Kent *et al*, 2010). As reference, the same analysis was conducted for all annotated lncRNAs and protein-coding genes (taken from Ensembl version 82). A more positive score indicates stronger conservation across various species.

## Animal experiments

All animal studies were approved by the Weizmann Institute's Animal Care and Use Committee. MDA-MB-231 cells intrinsically

labeled with RFP and expressing LIMT or eGFP (as control) and shLIMT or shControl where used to assess tumor growth and metastasis. For assessment of metastasis formation, 150,000 cells were injected into the tail vein of 5-week-old female SCID mice. Eighteen days after injection, lungs were excised and imaged. The number of metastatic nodules in each lung was calculated using the Fiji software. For randomization, the weights of all mice were determined prior to injections and mice were separated into groups such that the average weight of each group was similar (average weight 18.2 ± 1.5 g). In order to conduct blind analyses, images of the excised lung were numbered by an uninvolved scientist. Animal identity was disclosed only after quantification of the number of metastatic nodules was made. Samples that fell outside the 1.5× interquartile range were considered as outliers and were omitted from further analysis.

### Clinical datasets

Three datasets were used. (i) METABRIC (Curtis *et al*, 2012), which contains gene expression profiles of ∼2,000 breast cancer patients. Profiling was done using Illumina's gene expression microarrays and survival of patients was followed for up to 25 years from diagnosis; (ii) a dataset (Gyorffy *et al*, 2010) that analyzed patient survival (www.kmplot.com) based on gene expression and clinical data from Gene Expression Omnibus (GEO). The version of the online tool we used analyzed 1,660 breast cancer patients; and (iii) Oslo2, a consecutive study that collects specimens from breast cancer patients referred to primary surgical treatment in the Oslo (Norway) region (Aure *et al*, 2014). For Oslo2, expression was measured using SurePrint G3 Human GE 8x60K one-color microarrays (Agilent Technologies), according to the manufacturer's protocol. For each sample, RNA (100 ng) was amplified and hybridized to an array, which included 42,405 unique 60-mer probes, targeting 27,958 Entrez genes and 7,419 lncRNAs. Scanning was performed with an Agilent Scanner G2565A. Signals were extracted using Feature Extraction v.10.7.3.1 (Agilent Technologies). Arrays were log2-transformed, normalized, and hospital-adjusted by subtracting from each probe value the median probe value among samples from the same hospital. Probes with identical Entrez ID were averaged to form a single expression value per gene. Molecular subtypes of disease were derived using the PAM50 classifier (Parker *et al*, 2009). Subtypes were available for 381 tumors. To evaluate differences in lncRNA expression between two groups, a Student's *t*-test was applied, and to evaluate differences in expression among three or more groups, a one-way analysis of variance (ANOVA) was applied. Data from 309 tumors were available for comparison of lncRNA expression between different clinical subgroups.

### Patient survival analyses

Using the METABRIC dataset, the cohort was divided for each probe into three groups of identical sizes, according to expression level of the measured lncRNA. A *P*-value was calculated for the difference in survival of the highest expressing group, compared to the lowest expressing group. Using the KM-plotter dataset, high- and low-expressing groups were defined by the median expression of each probe in the cohort. In both cases, *P*-values were corrected for

**The paper explained**

**Problem**
Breast cancer is a highly heterogeneous disease, the classification and understanding of which is still incomplete. Further in-depth analysis of both protein-coding and noncoding genes is vital to accelerate the resolution of these issues.

**Results**
Because growth factors play essential roles in both mammary gland development and in breast cancer progression, and long noncoding RNAs (lncRNAs) are emerging as cardinal regulators of gene expression, we profiled epidermal growth factor (EGF) inducible expression of lncRNAs in normal mammary cells. A subset of the EGF-inducible lncRNAs we identified correlated with clinical outcome of breast cancer patients. One of these, a highly conserved lncRNAs we called LIMT, is downregulated by EGF due to histone deacetylation at the respective promoter. In agreement with this, LIMT is expressed at low levels in the basal-like and in the HER2-enriched, two relatively aggressive subtypes of breast cancer. Furthermore, LIMT was found to inhibit the migration of mammary cells and to reduce metastasis formation *in vivo*.

**Impact**
In general, our results confirm and extend the roles of lncRNAs in progression of breast cancer. Specifically, we identify LIMT as a marker and putative driver of two relatively aggressive subtypes of breast cancer.

multiple hypotheses testing (Bonferroni) according to the number of probes tested in the relevant dataset.

### Statistical analysis

The statistical analysis of each assay appears in the relevant figure legend. For Student's *t*-test, unless otherwise indicated, the test was conducted as a two-sided unpaired test. Bonferroni correction was used to adjust *P*-values for multiple hypotheses in relevant cases. Error bars represent standard deviations. All experiments were carried out in triplicates, unless specified otherwise.

**Expanded View** for this article is available online.

### Acknowledgements

We thank members of our groups for support and help. We thank Dr. Swati Srivastava for her creative contribution. Our research is supported by the U.S. National Cancer Institute (Grant 5R37CA072981), the European Research Council, the Seventh Framework Program of the European Commission, the German-Israeli Project Cooperation (DIP), the Israel Cancer Research Fund, the Rising Tide Foundation, the Israel Science Foundation (ISF), and the Dr. Miriam and Sheldon G. Adelson Medical Research Foundation. Y.Y. is the incumbent of the Harold and Zelda Goldenberg Professorial Chair in Molecular Cell Biology. L.L. was partially supported by Israel Ministry of Science and Technology and by an ISEF Fellowship. M.R.A was a postdoctoral fellow of the South-Eastern Norway Regional Health Authority (grant 368039-6051-39648). Research in the Diederichs laboratory is supported by the German Research Foundation (DFG Transregio TRR77, Di 1421/7-1), the Excellence Cluster CellNetworks, the Helmholtz Society, the Virtual Helmholtz Institute for Resistance in Leukemia and the European Union (Marie Curie Grant).

## Author contributions

AS-C and YY were involved in conception and design of the study, as well as writing the paper. AS-C, SC, YE, SL, JY, NN, YK, and CK were involved in acquisition of data. AS-C, MRA, LL, YE, FA, and IU were involved in analysis and interpretation of data. MP-S and SD provided RNA from human cell lines. OSBREAC, ALBD, and VNK provided clinical data. YY, ZY, VNK, SW, and A-LB-D supervised research and analyses.

## Conflict of interest

The authors declare that they have no conflict of interest.

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

**The Oslo Breast Cancer Consortium (OSBREAC)**

Torill Sauer[1,2], Jürgen Geisler[3,4,5], Solveig Hofvind[6,7], Tone F Bathen[8], Elin Borgen[9], Anne-Lise Børresen-Dale (listed as author)[4,10,11], Olav Engebråten[12,13,14], Øystein Fodstad[12,14], Øystein Garred[15], Gry Aarum Geitvik[10,11], Rolf Kåresen[16,17], Bjørn Naume[4,13], Gunhild Mari Mælandsmo[12,18], Hege G Russnes[10,15], Ellen Schlichting[19], Therese Sørlie[14], Ole Christian Lingjærde[20,21], Vessela N Kristensen (listed as author)[4,5,22], Kristine Kleivi Sahlberg[23,24], Helle Kristine Skjerven[25], Britt Fritzman[26]

[1]Department of Pathology, Akershus University Hospital, Lørenskog, Norway. [2]Institute of Clinical Medicine, Faculty of Medicine, University of Oslo, Oslo, Norway. [3]Department of Oncology, Akershus University Hospital, Lørenskog, Norway. [4]Institute for Clinical Medicine, University of Oslo, Oslo, Norway. [5]Division of Medicine, Akershus University Hospital, Lørenskog, Norway. [6]Cancer Registry of Norway, Oslo, Norway. [7]Oslo and Akershus University College of Applied Sciences, Faculty of Health Science, Oslo, Norway. [8]Department of Circulation and Medical Imaging, Norwegian University of Science and Technology (NTNU), Trondheim, Norway. [9]Department of Pathology, Division of Diagnostics and Intervention, Oslo University Hospital, Oslo, Norway. [10]Department of Cancer Genetics, Institute for Cancer Research, Oslo University Hospital. [11]The Norwegian Radium Hospital, Oslo, Norway. [12]Department of Tumor Biology, Institute for Cancer Research, Oslo University Hospital, Oslo, Norway. [13]Department of Oncology, Division of Surgery and Cancer and Transplantation Medicine, Oslo University Hospital, Oslo, Norway. [14]Institute for Clinical Medicine, Faculty of Medicine, University of Oslo, Oslo, Norway. [15]Department of Pathology, Oslo University Hospital, Oslo, Norway. [16]Institute of Clinical Medicine, University of Oslo, Oslo, Norway. [17]Department of Breast- and Endocrine Surgery, Division of Surgery, Cancer and Transplantation, Oslo University Hospital, Oslo, Norway. [18]Department of Pharmacy, Faculty of Health Sciences, University of Tromsø, Tromsø, Norway. [19]Section for Breast- and Endocrine Surgery, Department of Cancer, Division of Surgery, Cancer and Transplantation Medicine, Oslo University Hospital, Oslo, Norway. [20]Centre for Cancer Biomedicine, University of Oslo, Oslo, Norway. [21]Department of Computer Science, University of Oslo, Oslo, Norway. [22]Department of Clinical Molecular Biology and Laboratory Science (EpiGen). [23]Department of Research, Vestre Viken Hospital, Drammen, Norway. [24]Department of Cancer Genetics, Institute for Cancer Research, Oslo University Hospital Radiumhospitalet, Norway. [25]Breast and Endocrine Surgery, Department of Breast and Endocrine Surgery, Vestre Viken Hospital, Drammen, Norway. [26]Østfold Hospital, Østfold, Norway.

