## [Review Process File · EMBO Molecular Medicine]

LIMT is a novel metastasis inhibiting lncRNA suppressed by EGF and downregulated in aggressive breast cancer

Aldema Sas-Chen, Miriam R. Aure, Limor Leibovich, Silvia Carvalho, Yehoshua Enuka, Cindy Körner, Maria Polycarpou-Schwarz, Sara Lavi, Nava Nevo, Yuri Kuznetsov, Justin Yuan, Francisco Azuaje, Oslo Breast Cancer Research Consortium (OSBREAC), Igor Ulitsky, Sven Diederichs, Stefan Wiemann, Zohar Yakhini, Vessela N. Kristensen, Anne-Lise Børresen-Dale and Yosef Yarden

Corresponding author: Yosef Yarden, The Weizmann Institute of Science

Review timeline:	Submission date:	08 January 2016
	Editorial Decision:	16 February 2016
	Revision received:	16 May 2016
	Editorial Decision:	01 June 2016
	Revision received:	09 June 2016
	Accepted:	24 June 2016

Transaction Report:

Editor: Roberto Buccione

1st Editorial Decision

16 February 2016

Thank you for the submission of your manuscript to EMBO Molecular Medicine. We have now heard back from the three Reviewers whom we asked to evaluate your manuscript.

Please accept our apologies for the unusual delay, due also to further discussion and consulting with an additional advisor.

Although reviewers 1 and 3 are globally positive, reviewer 2 is much less so. Specifically, the latter feels that the functional data are not convincing and also laments the lack of mechanistic insight on how LIMT is induced or exerts its functions. Reviewer 2 also notes that the manuscript is not well written. Thus, s/he raises very important issues but admittedly offers very little in terms of explanation and is a bit cursory. During our reviewer cross-commenting process, reviewer 2 confirmed his/her doubts, whereas another disagreed that provision of mechanistic insight was fundamental for publication. Although I tend to agree in this case that such insight would not be an absolute requirement, at least the phenotype reported must be very solid and on this Reviewer 2 expressed concerns as well; I'm afraid I agree.

To collect an additional independent opinion, I sought the advice of an external expert, who was not immediately available, hence the further delay. The expert agreed that provision of further mechanistic insight would not be an absolute requirement in this case given the finding of a new suppressive lncRNA. S/he did fully agree however, that the quality of the data needs to be

significantly improved and specifically found the in vitro migration effects to be modest and unconvincing, especially with the KD, and felt the in vivo effects to be only marginally more convincing and based on lncRNA overexpression and thus rather un-physiological. The advisor suggested a key experiment that would go a long way in that respect. S/he suggests to CRISPR-out the lncRNA from the MCF10A cells and then use the KO cells to repeat the in vitro and in vivo experiments, including 'rescue' experiments by adding back the LIMT. This would circumvent that fact that siRNAs, especially those against lncRNAs, tend to be highly prone to off-target effects that might mislead. It was also noted that Fig. 3e has no loading controls and as such is not very meaningful. I should add that the advisor also mentioned that the manuscript needs a major stylistic overhaul and found it "surprisingly poorly written", hence agreeing with reviewer 2.

In conclusion, while publication of the paper cannot be considered at this stage, given the potential interest of your findings and after internal discussion, we have decided to give you the opportunity to address the criticisms along the lines suggested by our advisor. If the results from these easy to perform and not overly time consuming experiments confirmed your conclusions, then I would forego the request for further mechanistic insight.

Please note that it is EMBO Molecular Medicine policy to allow a single round of revision only and that, therefore, acceptance or rejection of the manuscript will depend on the completeness of your responses included in the next, final version of the manuscript.

EMBO Molecular Medicine now requires a complete author checklist (<http://embomolmed.embopress.org/authorguide#editorial3>) to be submitted with all revised manuscripts. Provision of the author checklist is mandatory at revision stage; The checklist is designed to enhance and standardize reporting of key information in research papers and to support reanalysis and repetition of experiments by the community. The list covers key information for figure panels and captions and focuses on statistics, the reporting of reagents, animal models and human subject-derived data, as well as guidance to optimise data accessibility.

As you know, EMBO Molecular Medicine has a "scooping protection" policy, whereby similar findings that are published by others during review or revision are not a criterion for rejection. However, I do ask you to get in touch with us after three months if you have not completed your revision, to update us on the status. Please also contact us as soon as possible if similar work is published elsewhere.

I look forward to seeing a revised form of your manuscript as soon as possible.

***** Reviewer's comments *****

Referee #1 (Comments on Novelty/Model System):

This manuscript reports phenomenologically on the finding of lncRNAs and puts these into context of erbB1 and erbB2. It is postulated that the finding of lncRNAs (LINC01089) is depleted in triple-negative and Her-2/neu positive tumors and constitutes a prognostic indicator. It is concluded that lncRNAs might be important as prognostic biomarkers due to their ability to influence tumor cell progression.

Although very much of interest as a finding, the fact that the manuscript takes a rather phenomenological approach rather than an explanatory one, the content should be driven into the direction of an attempt to understand the underlying mechanisms resulting in the described phenomena. This would contribute to our understanding of the underlying biological processes rather than their phenomenologic (although very valid!) prognostic context.

Referee #2 (Comments on Novelty/Model System):

This is a poorly written study with minimal insight into how this noncoding RNA species mediates these pleiotropic effects

Referee #2 (Remarks):

The paper is not well written and repetitive and would benefit from some editing

The data in Figure 3d AND 3F and 4D are not convincing and would benefit from repeat studies etc.

Another major deficit of the paper is the absence of definitive mechanistic insight of how this mRNA species is induced or functions. Collectively the paper is not yet ready for publication since it provides little or no mechanistic insight.

Referee #3 (Comments on Novelty/Model System):

This was a well-conceived and conducted study in an important area. The findings were novel and could have an impact on patient classification and treatment.

Referee #3 (Remarks):

The overall aim of this study was to identify EGF-regulated lncRNAs in breast cancer and to determine the prognostic significance and functional relevance of those identified. The authors identified a total of 346 EGF-regulated lncRNAs, most of which were down-regulated. They next looked at the prognostic relevance of these lncRNAs in two independent databases and identified 11. Among these was LINC01089 (LIMT) whose expression is reduced by EGF and whose reduced expression levels correlate with shortened overall and relapse-free survival. They carried out both knockdown and over-expression studies to demonstrate that LIMT plays a role as an inhibitor of metastasis and invasion in vitro and in vivo. They further showed that LIMT is regulated via the RAS to ERK pathway and that it is epigenetically regulated. Overall this was a well-written manuscript describing a carefully planned and executed study. It is acceptable for publication with only one minor revision:

1) Please explain why a 1.5-fold change was used for the cut-off to identify differentially expressed lncRNAs, but a 2-fold change was the cutoff for mRNAs.

1st Revision - authors' response

16 May 2016

Rebuttal Letter

Manuscript EMM-2016-06198 ("*LIMT is a novel metastasis inhibiting lncRNA suppressed by EGF and down-regulated in aggressive breast cancer*"; Aldema Sas-Chen et al.)

PART 1: Responses to Editor's comments

Although reviewers 1 and 3 are globally positive, reviewer 2 is much less so. Specifically, the latter feels that the functional data are not convincing and also laments the lack of mechanistic insight on how LIMT is induced or exerts its functions.

In response to this comment, we added to the revised manuscript the following new evidence:

(i) Functional data: We collected gene expression data from LIMT-manipulated cells, both LIMT-overexpressing and LIMT-knockdown cells (see our response to Referee 1). Amongst the altered genes, we identified some that regulate cell migration, raising the possibility that LIMT affects the respective promoters.

(ii) Mechanistic insights: We included in the revised manuscript improved experimental data showing that induction of LIMT is regulated through activation of the ERK-MAPK cascade (see our response to Referee 2). In addition, our revision better highlights data relevant to the mechanism of LIMT's promoter regulation due to EGF-induced and ERK-mediated reduced acetylation of histone 3 (H3K27Ac) at the transcription start site (TSS) of *LIMT*.

(iii) RNA-protein interactions: Assuming that LIMT directly interacts with specific proteins, we conducted an RNA affinity purification (RAP) assay, whereby biotin-labeled LIMT transcripts (or control transcripts) were used to pull down proteins from cell extracts (see our response to Referee 1). This identified several candidates, including a known RNA-binding protein implicated in cell migration. However, validation of specific binders and their mode of action in migration and metastasis will require 10-15 more months of intense work.

Reviewer 2 also notes that the manuscript is not well written.

In response to this comment, the manuscript has been revised and several portions of the text were re-written. We note, however, that Referee 3 indicated the following in his/her critique: "Overall this was a *well-written* manuscript describing a carefully planned and executed study".

Thus, s/he raises very important issues but admittedly offers very little in terms of explanation and is a bit cursory. During our reviewer cross-commenting process, reviewer 2 confirmed his/her doubts, whereas another disagreed that provision of mechanistic insight was fundamental for publication. Although I tend to agree in this case that such insight would not be an absolute requirement, at least the phenotype reported must be very solid and on this Reviewer 2 expressed concerns as well; I'm afraid I agree.

To collect an additional independent opinion, I sought the advice of an external expert, who was not immediately available, hence the further delay. The expert agreed that provision of further mechanistic insight would not be an absolute requirement in this case given the finding of a new suppressive lncRNA. S/he did fully agree however, that the quality of the data needs to be significantly improved and specifically found the *in vitro* migration effects to be modest and unconvincing, especially with the KD, and felt the *in vivo* effects to be only marginally more convincing and based on lncRNA overexpression and thus rather un-physiological.

In response to these comments and in order to further support the phenotype of LIMT, we first established clones of human mammary cells stably expressing relatively low levels of LIMT. This was achieved by designing several different shRNAs and selecting the two most effective ones (denoted: sh716 and sh1483). Once in hand, the selected shRNAs were employed both *in vitro* and in animal studies (see our response to Referee 2). Although the two shRNAs target different parts of the LIMT transcript, they exhibited similar results both *in vitro* and *in vivo*. These results were in line with the observations made by the siRNA experiments. In fact, the use of shRNAs resulted in stronger effects than those seen with siRNAs, a fact that can be attributed to the more robust knockdown achieved by stable expression rather than a transient one. The higher quality results of the *in vitro* migration assays, as well as the new observations we made in animals, further confirm the phenotype of LIMT. In addition, the new results answer the Referees' comments because they are based on knockdown rather than overexpression studies.

The advisor suggested a key experiment that would go a long way in that respect. S/he suggests to CRISPR-out the lncRNA from the MCF10A cells and then use the KO cells to repeat the *in vitro* and *in vivo* experiments, including 'rescue' experiments by adding back the LIMT. This would circumvent that fact that siRNAs, especially those against lncRNAs, tend to be highly prone to off-target effects that might mislead. It was also noted that Fig. 3e has no loading controls and as such is not very meaningful. I should add that the advisor also mentioned that the manuscript needs a major stylistic overhaul and found it "surprisingly poorly written", hence agreeing with reviewer 2.

- (a) Using CRISPR to address functions of LIMT: As requested, we attempted to CRISPR-out *LIMT* from MCF10A cells and then use the KO cells to repeat the *in vitro* and *in vivo* experiments, including 'rescue' experiments. As a first step, we designed twelve single-guide RNAs (sgRNAs) directed against different locations in the promoter and in the transcript area of LIMT using the design tool available at the Zhang Lab website (<http://www.genome->

engineering.org/crispr/?page_id=41) (see Figures I, below). Each sgRNA was sub-cloned into the pX330 plasmid, which also contained the Cas9 gene. We then transiently transfected MCF10A cells with the sgRNA-containing plasmid and an eGFP-expressing plasmid. Co-transfection with eGFP permitted sorting specific cells (using FACS), twenty-four hours after transfection. Cells were then seeded as single cells in 96-well plates followed by an expansion period of at least four weeks, to allow single cell clones reach a sufficiently large size for DNA profiling. Once DNA was extracted from cells, PCR was performed using primers spanning the respective deleted regions in order to detect positive clones.

Unfortunately, none of the clones we screened at this stage showed any of the expected deletions. To overcome this, we switched to an alternative plasmid system, namely: lentiCas9-Blast (Addgene ID: #52962) and lentiGuide-Puro (Addgene ID: #52963). This system encodes selectable markers, as well as the Cas9 open reading frame and specific sgRNAs. We then produced lentiviruses encapsulating either Cas9 or the sgRNA plasmids in HEK-293FT cells, and used them to infect MCF10A cells (for in vitro assays) or RFP-labeled MDA-MB-231 cells (for experiments using animals). Selection of cells was done using puromycin and blasticidin. Whenever mock-infected cells exhibited ultimate death by the antibiotics, the remaining cells from the Cas9-sgRNA infected cells were seeded in serial dilutions and maintained for expansion for at least 4 weeks (as single cell colonies). We then screened DNA from different clones, using the PCR technique mentioned above. A few of the clones exhibited DNA bands that corresponded to deleted versions of the *LIMT* gene (Fig. I-C). These bands were extracted from the agarose gel and a deletion was verified by Sanger sequencing (Fig. I-D). As seen in Fig. I-C the clones showing the deletion also showed additional bands, some of which corresponded to the WT sequence of the gene. This might result from two reasons: first, it could be that only one allele of the clone was deleted and thus WT sequences are still present. Second, it could be that the clones we picked are not pure, and actually consist of a few types of cells. For these reasons we re-plated the positive clones at low density to re-pick single clones. This process is still ongoing as re-plated cells need at least 4 weeks to reach an appropriate size for DNA profiling. We estimate that completion of all planned in vitro and animal studies will require additional 5-8 months.

(c) A major stylistic overhaul: As requested by two Referees, we revised the whole text and introduced several stylistic corrections.

Figure II. LIMT is induced downstream to the ERK signaling pathway. MCF10A cells were treated for 30 minutes with U0126 (a MEK inhibitor), or with no agent, and thereafter EGF was added (10 ng/ml) and incubated with cells for up to four hours. Thirty minutes after the addition of EGF, cells were lysed and analyzed, using immunoblotting, for the levels of phosphorylated ERK, along with the level of total ERK (loading control).

Note: This figure was included in the revised manuscript (Fig. 3C).

In conclusion, while publication of the paper cannot be considered at this stage, given the potential interest of your findings and after internal discussion, we have decided to give you the opportunity to address the criticisms along the lines suggested by our advisor. If the results from these easy to perform and not overly time consuming experiments confirmed your conclusions, then I would forego the request for further mechanistic insight.

PART 2: Responses to Referee 1's comments

This manuscript reports phenomenologically on the finding of lncRNAs and puts these into context of erbB1 and erbB2. It is postulated that the finding of lncRNAs (LINC01089) is depleted in triple-negative and Her-2/neu positive tumors and constitutes a prognostic indicator. It is concluded that lncRNAs might be important as prognostic biomarkers due to their ability to influence tumor cell progression.

Although very much of interest as a finding, the fact that the manuscript takes a rather phenomenological approach rather than an explanatory one, the content should be driven into the direction of an attempt to understand the underlying mechanisms resulting in the described phenomena. This would contribute to our understanding of the underlying biological processes rather than their phenomenologic (although very valid!) prognostic context.

In order to more deeply dissect the mechanism of action of LIMT, we manipulated its level of expression and searched for genes it might regulate. Thus, we used the following two derivatives of MCF10A cells, along with the parental subline: (i) a clone that stably overexpresses LIMT, and (ii) a clone in which LIMT was knocked-down by using specific siRNAs. Next, we isolated RNA from the three types of cells and profiled expression of both coding and non-coding genes using Affymetrix microarrays. The results we obtained are shown below (Figure III). Note that Figure III presents genes that were affected by the two opposing genetic manipulations and exerted a statistically significant change of at least 1.5-fold (relative to control) in at least one of the interventions (adjusted p value < 0.05, triplicates of all samples). Highlighted are genes known to regulate cancer progression and/or cellular migration. How exactly LIMT regulates expression of these presumed target genes is a matter for future research.

Figure III. Manipulation of *LIMIT* expression affects the abundance of several transcripts involved in cell migration and in tumor progression. *LIMIT* was stably overexpressed or knocked-down (by using siRNAs) in MCF10A cells and levels of the lncRNA were assessed by real-time qPCR (data not shown). In the overexpression experiments, transfection with eGFP was used as control. Affymetrix oligonucleotide arrays were used to survey differential gene expression patterns. Shown are 48 genes that showed a significant difference of at least 1.5-fold under at least one of the conditions. Genes are presented in two clusters according to their response to manipulation of *LIMIT* (OX, overexpression; KD, knock-down). Highlighted are genes previously implicated in cancer progression and/or regulation of cell migration, along with genes displaying significant correlation between expression levels and clinical outcome of breast cancer patients (according to the METABRIC dataset).

Note: This figure was included in the revised manuscript (Figure EV5).

PART 3: Responses to Referee 2's comments

The paper is not well written and repetitive and would benefit from some editing

The data in Figure 3d AND 3F and 4D are not convincing and would benefit from repeat studies etc.

Another major deficit of the paper is the absence of definitive mechanistic insight of how this mRNA species is induced or functions. Collectively the paper is not yet ready for publication since it provides little or no mechanistic insight.

- (i) Figure 3D: In response to these comments we revised all three figures indicated by the Referee. The revised version of Figure 3D shows results obtained using MCF10A cells, in which we either knocked-down *LIMIT*, using shRNAs, or overexpressed the lncRNA. Both migration and invasion assays were performed after we validated, using qPCR, the altered levels of *LIMIT* expression. Note that the shRNA we used was selected out of a group of ten, and used to establish a stable clone of MCF10A cells. Figure IV (below) presents two shRNAs, sh716 and sh1483, and their effects on *LIMIT* expression. On the basis of the results, we

selected sh716 for further experiments. Figure V exemplifies and quantifies the functional assays we performed using siRNAs, sh716, and overexpression of LIMT. The results we obtained by utilizing all three independent approaches consistently confirmed the ability of LIMT to inhibit both migration and invasion of human mammary cells.

Figure IV. Stable knockdown of LIMT in MCF10A and MDA-MB-231 cells. (A) A scheme depicting the location of two shRNAs designed to knockdown the expression of LIMT in cells. (B) RNA was extracted from MCF10A and MDA-MB-231 cells stably expressing shRNAs against LIMT, or a control shRNA. The isolated RNA was converted into cDNA and expression of LIMT was measured using RT-qPCR.

Note: This figure was not included in the revised manuscript.

Figure V. LIMT is a negative regulator of mammary cell migration and invasion. (A) Migration assays were conducted with MCF10A cells transiently transfected with LIMT-specific (or control) oligonucleotides. Alternatively, we employed MCF10A cells stably expressing an shRNA against LIMT (sh716; or a control shRNA), or cells overexpressing the *LIMT* gene (or eGFP as control). Quantification of cell migration is presented (mean \pm S.D). (B) Invasion assays were conducted with MCF10A cells pre-treated as in A. Quantification of invasion activity is presented as means \pm S.D.

Note: This figure was included in the revised manuscript (Figure 3A and Figure 3B).

(ii) **Figure 3F:** As requested, we have repeated the experiments showing the effect of inhibiting MEK on *LIMT* expression. The new figure is shown below (Fig. VI). Importantly, the data present results obtained in all four experiments we carried out. Notably, all experiments support the same conclusion, namely: blocking the ERK pathway prevented the ability of EGF to induce downregulation of *LIMT*.

Figure VI. Downregulation of *LIMT* expression is abolished when the ERK-MAPK pathway is inhibited using a MEK inhibitor. MCF10A cells were treated for 30 minutes with U0126 (a MEK inhibitor), or with no agent, and thereafter EGF was added (10 ng/ml) and incubated with cells for up to four hours. (A) Thirty-minutes after the addition of EGF, cells were lysed and analyzed, using immunoblotting, for the levels of phosphorylated ERK (and total ERK as a loading control). (B) RNA was purified after 4 hours of EGF stimulation and used to determine levels of LIMT transcripts using real-time qPCR. Expression values of four experiments are presented as log₂ fold of the change relative to time zero of each condition, after normalization to beta-2-microglobulin.

Note: This figure was included in the revised manuscript (Figures 3C and 3D).

(iii) **Figure 4D:** As requested by Referee 2, we extended the analyses of metastasis formation. Importantly, in the revised manuscript we undertook an shRNA strategy, which complemented LIMT overexpression. To this end, we designed 10 different shRNA sequences, cloned each into a lentiviral plasmid and separately infected both MDA-MB-231 and MCF10A cells. Next, we selected a clone that showed maximal knockdown of LIMT expression (as measured by using qPCR) and used the corresponding MDA-MB-231 cells for metastasis assays, which employed injection of cells into the tail vein of immunocompromised mice. The results of this experiment are

presented in panel A of the figure shown below (Figure VII). Clearly, shLIMT-treated cells more effectively colonized the lungs of the mice we treated, in line with a role played by LIMT in metastasis formation. The same protocol was used to test LIMT-overexpressing cells, and although the statistical power of the overexpression strategy was weaker (p value of 0.02; student's t -test) than the knockdown strategy, both approaches lead us to conclude that LIMT inhibits metastasis of mammary cells in our animal models.

Figure VII. LIMT inhibits metastasis formation in vivo. (A) MDA-MB-231 cells stably expressing shRNAs against LIMT or control shRNAs were injected into the tail vein of 5-weeks old female SCID mice (150,000 cells/mouse). Eighteen days after injection, lungs were excised and imaged. The number of metastatic nodules in each lung was quantified and presented in a dot plot. The horizontal lines represent median number of nodules per animal of each group. Shown are representative fluorescence images of lungs from the LIMT knockdown group (N=7) and the control group (N=8). (B) The experiment described above was carried out with MDA-MB-231 cells overexpressing LIMT or eGFP. Shown are representative fluorescence images of lungs from the LIMT-overexpressing group (N=11) and the control group (N=7).

Note: This figure was included in the revised manuscript (Figure 4).

(iv) Mechanistic insights: To address mechanisms underlying the cellular action of LIMT we searched for gene targets (see our response to Referee 1), as well as for putative protein targets. The latter endeavor entailed RNA affinity purification (see Figure VIII). Firstly, we transcribed LIMT *in vitro* while labeling the transcript with a biotinylated UTP. Next, we incubated beads decorated with the biotinylated LIMT with whole extracts of MCF10A cells and bound proteins were separated on a polyacrylamide gel. Silver staining of the gel revealed some potential protein partners of LIMT, which were cut from the respective lane and analyzed using mass-spectrometry (MS). MS analysis revealed enrichment for several proteins (especially in the 70 kilodalton region). However, reciprocal experiments that employed immunoprecipitation of the identified proteins followed by PCR aimed (aiming at identifying LIMT transcripts) have so far failed. Additional control experiments and analyses of more protein bands are underway.

Figure VIII. LIMT might interact with the RNA-binding proteins. (A) A scheme of the RNA affinity purification assay conducted in order to identify proteins bound to the LIMT transcript. (B) Protein extracts from MCF10A cells were incubated with *in vitro* transcribed RNA of LIMT (or a control lncRNA, LOC388796), which were either labeled or not with biotinylated UTPs. LncRNA-bound proteins were then separated on a polyacrylamide gel and stained using a silver nitrate solution. The region corresponding to proteins at the size of 70 kDa was cut from both biotin-positive lanes and analyzed using mass-spectrometry (MS). The experiment was performed in four biological replicates. Two of the replicates were used for MS analysis.

Note: This figure was not included in the revised manuscript.

PART 4: Responses to Referee 3's comments

The overall aim of this study was to identify EGF-regulated lncRNAs in breast cancer and to determine the prognostic significance and functional relevance of those identified. The authors identified a total of 346 EGF-regulated lncRNAs, most of which were down-regulated. They next looked at the prognostic relevance of these lncRNAs in two independent databases and identified 11. Among these was LINC01089 (LIMT) whose expression is reduced by EGF and whose reduced expression levels correlate with shorted overall and relapse-free survival. They carried out both knockdown and over-expression studies to demonstrate that LIMT plays a role as an inhibitor of metastasis and invasion in vitro and in vivo. They further showed that LIMT is regulated via the RAS to ERK pathway and that it is epigenetically regulated. Overall this was a well-written manuscript describing a carefully planned and executed study. It is acceptable for publication with only one minor revision:

1) Please explain why a 1.5-fold change was used for the cut-off to identify differentially expressed lncRNAs, but a 2-fold change was the cutoff for mRNAs.

The paragraph below and Figure IX showing the range of RNA variations address the question of the Referee.

Our analysis of the microarray data, which surveyed expression of lncRNAs and mRNAs in response to EGF stimulation, revealed that the overall dynamic range of lncRNAs was smaller than the dynamic range corresponding to mRNAs (i.e., the values of fold change ranged from -3.67 to 10.36 for mRNAs and from -2.32 to 6.94 for lncRNAs). Due to the narrower lncRNA's dynamic range, transcripts were considered dynamic if their fold-change was greater than 1.5 (lncRNAs) or 2 (mRNAs), relative to time zero, in at least one time point. This explanation was added to the revised manuscript.

Figure IX. Non-coding RNAs exhibit a smaller dynamic range of expression compared to protein-coding RNAs. MCF10A cells were stimulated with EGF for increasing time periods (0, 20, 40, 60, 120, 240, 480 minutes). RNA was then extracted and used to assess expression of coding and non-coding RNAs (ncRNAs) using microarrays. For each transcript the maximal change in expression from time zero (no stimulation) was calculated. The graph shows the distribution of log₂ fold change of all transcripts, the expression of which changed by at least 0.6-fold (corresponding to a linear fold change of 1.5, as reported in the manuscript). Overall, lncRNAs presented a smaller range of change in expression.

Note: This figure was not included in the revised manuscript.

2nd Editorial Decision

01 June 2016

Thank you for the submission of your revised manuscript to EMBO Molecular Medicine. We have now received the enclosed reports from the referees that were asked to re-assess it. As you will see the reviewers are now globally supportive and I am pleased to inform you that we will be able to accept your manuscript pending the following final amendments:

- 1) While performing our pre-publishing quality control and image screening routines, we noticed in Fig. 4A, a vertical discontinuity towards the left third in the both the eGFP and LIMT image panels. Please explain and provide better images.
- 2) I have slightly edited the Abstract and "The paper explained" sections (please see attached manuscript). Please accept/modify as appropriate using the attached version.
- 3) We encourage the publication of source data, particularly for electrophoretic gels and blots, with the aim of making primary data more accessible and transparent to the reader. Would you be willing to provide a PDF file per figure that contains the original, uncropped and unprocessed scans of all or at least the key gels used in the manuscript? The PDF files should be labeled with the appropriate figure/panel number, and should have molecular weight markers; further annotation may be useful but is not essential. The PDF files will be published online with the article as supplementary "Source Data" files. If you have any questions regarding this just contact me.
- 4) Data described in submitted manuscripts should be deposited in a MIAME-compliant format with one of the public databases. We would therefore ask you to submit your microarray data to the ArrayExpress database maintained by the European Bioinformatics Institute for example. ArrayExpress allows authors to submit their data to a confidential section of the database, where they can be put on hold until the time of publication of the corresponding manuscript. Please see <http://www.ebi.ac.uk/arrayexpress/Submissions/> or contact the support team at arrayexpress@ebi.ac.uk for further information.

Please submit your revised manuscript within two weeks. I look forward to seeing a revised form of your manuscript as soon as possible.

***** Reviewer's comments *****

Referee #1 (Comments on Novelty/Model System):

From my point, all relevant remarks and criticism were answered.

Referee #2 (Comments on Novelty/Model System):

I think this paper is now ready for publication and believe they have answered all of my issues

2nd Revision - authors' response

09 June 2016

Thank you for the opportunity to correct and enhance the manuscript (EMM-2016-06198). Below we refer to all points indicated in your acceptance letter of June 1st, 2016. Because both Referees had no specific comments, we only relate to the editorial points.

1) While performing our pre-publishing quality control and image screening routines, we noticed in Fig. 4A, a vertical discontinuity towards the left third in the both the eGFP and LIMT image panels. Please explain and provide better images.

The vertical line appearing in the images of the lungs shown in Figure 4A is apparently a machine made line incorporated by the equipment used to acquire images from animals. This faint line becomes visible upon image enhancement, which is necessary for visualization of fluorescent metastatic nodules. Notably, the line is more visible in images with a low metastatic count. Unfortunately, all images taken in the experiment show the same line. Note that we deposited the raw files of the metastasis images as "Source Data files". Of course we will collaborate with your Production Editor to minimize the issue.

2) I have slightly edited the Abstract and "The paper explained" sections (please see attached manuscript). Please accept/modify as appropriate using the attached version.

All editorial corrections were integrated into the revised version of the manuscript.

3) We encourage the publication of source data, particularly for electrophoretic gels and blots, with the aim of making primary data more accessible and transparent to the reader. Would you be willing to provide a PDF file per figure that contains the original, uncropped and unprocessed scans of all or at least the key gels used in the manuscript? The PDF files should be labeled with the appropriate figure/panel number, and should have molecular weight markers; further annotation may be useful but is not essential. The PDF files will be published online with the article as supplementary "Source Data" files. If you have any questions regarding this just contact me.

As requested, we have submitted a "Source Data file" with the full blot shown in Figure 3 of the manuscript. The corresponding PDF file is shown below.

The ERK pathway mediates the effect of EGF on LIMT, which normally inhibits mammary cell migration and invasion. MCF10A cells were treated for 30 minutes with U0126 (a MEK inhibitor), or with no agent, and thereafter EGF was added (10 ng/ml) and incubated with cells for

additional 30 minutes. Protein extracts were used to assess levels of phosphorylated ERK (upper) and total level of ERK (bottom). The blot was used as the raw data for Figure 3C.

4) Data described in submitted manuscripts should be deposited in a MIAME-compliant format with one of the public databases. We would therefore ask you to submit your microarray data to the ArrayExpress database maintained by the European Bioinformatics Institute for example. ArrayExpress allows authors to submit their data to a confidential section of the database, where they can be put on hold until the time of publication of the corresponding manuscript. Please see <http://www.ebi.ac.uk/arrayexpress/Submissions/> or contact the support team at arrayexpress@ebi.ac.uk for further information.

As requested, we have uploaded all microarray data onto the ArrayExpress database. The revised text refers to the uploaded files as (i) E-MTAB-4821 (Affymetrix array quantifying changes in gene expression upon LIMT manipulation), and (ii) E-MTAB-4822 (Agilent array quantifying lncRNA expression after EGF stimulation).

Three final points:

- (i) We definitely agree that the Journal will upload our rebuttal letter, such that it will become widely available. Because the original letter referred to a potential binder of LIMT, a specific RNA-binding protein, which is the subject of our current analysis, we uploaded a new version of the rebuttal letter that does not indicate the name of the putative binder.
- (ii) Note that my correct Orcid number is as follows: 0000-0003-4168-7884
- (iii) We uploaded two files for the Synopsis of the manuscript (text and a figure). We annotated them as "Related Manuscript Files".

It is my hope that you will find the revised manuscript and all associated files suitable and correspondingly approve publication in EMBO Molecular Medicine.

Corresponding Author Name: Yosef Yarden

Manuscript Number: EMM-2016-06198